# Efficient Equivariant Network

**Lingshen He[1], Yuxuan Chen[1], Zhengyang Shen[3], Yiming Dong[1], Yisen Wang[1,2], Zhouchen Lin[1,2,4]***

[1]Key Laboratory of Machine Perception (MOE), School of Artificial Intelligence, Peking University
[2]Institute for Artificial Intelligence, Peking University
[3]School of Mathematical Sciences and LMAM, Peking University
[4]Pazhou Lab, Guangzhou 510330, China
`lingshenhe@pku.edu.cn, yuxuan_chen1997@outlook.com,`
`shenzhy@pku.edu.cn, yimingdong_ml@outlook.com,`
`yisen.wang@pku.edu.cn, zlin@pku.edu.cn`

## Abstract

Convolutional neural networks (CNNs) have dominated the field of Computer Vision and achieved great success due to their built-in translation equivariance. Group equivariant CNNs (G-CNNs) that incorporate more equivariance can significantly improve the performance of conventional CNNs. However, G-CNNs are faced with two major challenges: *spatial-agnostic problem* and *expensive computational cost*. In this work, we propose a general framework of previous equivariant models, which includes G-CNNs and equivariant self-attention layers as special cases. Under this framework, we explicitly decompose the feature aggregation operation into a kernel generator and an encoder, and decouple the spatial and extra geometric dimensions in the computation. Therefore, our filters are essentially dynamic rather than being spatial-agnostic. We further show that our *E*quivariant model is parameter *E*fficient and computational *E*fficient by complexity analysis, and also data *E*fficient by experiments, so we call our model $E^4$-Net. Extensive experiments verify that our model can significantly improve previous works with smaller model size. Especially, under the setting of training on $1/5$ data of CIFAR10, our model improves G-CNNs by $5\%+$ accuracy, while using only $56\%$ parameters and $68\%$ FLOPs.

## 1 Introduction

In the past few years, convolutional neural networks (CNNs) have been widely used and achieved superior results on multiple vision tasks, such as image classification [31, 55, 51, 22], semantic segmentation [3], and object detection [44]. A compelling explanation of the good performance of CNNs is that their built-in parameter sharing scheme brings in translation equivariance: shifting an image and then feeding it through a CNN layer is the same as feeding the original image and then shifting the resulted feature maps. In other words, the translation symmetry is preserved by each layer. Motivated by this, Cohen and Welling [9] proposed Group Equivariant CNNs (G-CNNs), showing how convolutional networks can be generalized to exploit larger groups of symmetries. Following G-CNNs, researchers have designed new neural networks that are equivariant to other transformations like rotations [9, 61, 24, 49] and scales [65, 53]. However, G-CNNs still have two main drawbacks: 1) In the implementation, G-CNNs would introduce extra dimensions to encode new transformations, such as rotations and scales, thus have a very high computational cost. 2) Although G-CNNs achieve group equivariance by sharing kernels, like vanilla CNNs, they lack the ability to adapt kernels to diverse feature patterns with respect to different spatial positions, namely, the spatial-agnostic problem [68, 39, 70, 71, 54, 67, 36].

---

*Corresponding author.

Some previous works focus on solving these two problems. Cheng et al. [4] proposed to decompose the convolutional filters over joint steerable bases to reduce model size. However, it is essentially G-CNNs which still have the inherent spatial-agnostic problem. To incorporate dynamic filters, one solution is introducing attention mechanism into each convolution layer in G-CNNs without disturbing inherent equivariance [48, 45]. The cost is that they introduce extra parameters and increase the complexity of space and time. Another solution is to replace group convolution layers with stand-alone self-attention layers by designing a specific position embedding to ensure equivariance [47, 26]. However, the self-attention mechanism suffers from quadratic memory and time complexity, because it has to compute the attention score at each pair of inputs.

Actually, Cohen et al. [7], Kondor et al. [29] and Bekkers [1] revealed that an equivariant linear layer is essentially a convolution-like operation. Inspired by this, we further discover that a general feature-extraction layer, either linear or non-linear, being equivariant is equivalent to that the feature aggregation mechanism between each pair of inputs only depends on the relative positions of these two inputs. Based on this observation, we propose a generalized framework of previous equivariant models, which includes G-CNNs and equivariant attention networks as special cases. Under this generalized framework, we design a new equivariant layer to conquer the aforementioned difficulties. Firstly, to avoid quadratic computational complexity, the feature aggregation operator is explicitly decomposed into a kernel generator and an encoder which takes one single feature as the input. Since our kernels are calculated based on input features, they are essentially dynamic rather than being spatial-agnostic. In addition, we decouple the feature aggregation mechanism across spatial and extra geometric dimensions to reduce the inter-channel redundancy in convolution filters [4] and further accelerate computation. Extensive experiments show that our method can process data very efficiently and perform significantly better than previous works using lower computational cost. As our method is parameter $E$fficient, computational $E$fficient, data $E$fficient and $E$quivariant, we name our new layer as $E^4$-layer.

We summarize our main contributions as follows:

- We propose a generalized framework of previous equivariant models, which includes G-CNNs and attention-based equivariant models as special cases.

- Under the generalized framework, we explicitly decompose the feature aggregation operator into a kernel generator and an encoder, and further decouple the spatial and extra geometric dimensions to reduce computation.

- Extensive experiments verify that our method is also data efficient and performs competitively with lower computational cost.

## 2   Related Work

Vanilla CNNs [34] are naturally translation equivariant. More symmetries are considered to be exploited into the network for different tasks, such as rotations over plane [9, 66, 35, 12, 61, 59, 49, 37, 52, 4, 41, 2, 10, 58, 21, 24], rotations over 3D space [62, 16, 57, 64, 14, 60, 15, 50, 28, 6], scaling [65, 40, 53, 46], symmetries on manifold [8, 11], and other general symmetry groups [17, 56, 18]. These works accomplish equivariance by constraining the linear mappings in layers, followed by pointwise non-linearities to enhance their expressive power. In general, researchers [29, 7, 1] pointed out that an equivariant linear mapping can always be written as a convolution-like integral, i.e., G-CNNs in practice. However, their theory is still limited to linear cases.

As works [68, 39, 70, 71, 54, 67, 36] point out the spatial-agnostic problem of CNNs and attention mechanisms [25, 63, 43, 13, 20] achieve impressive results on various vision tasks, researchers start to consider non-linear equivariant mapping. Romero et al. [48, 45] directly reweighted the convolution kernels with attention weights generated by features and obtained non-linear equivariant models. However, compared with G-CNNs, these methods introduce extra parameters and operations, resulting in an even heavier computational burden. Also, some works [47, 26, 23] proposed group equivariant self-attention [43, 13]. Fuchs et al. [19] incorporated self-attention into 3D equivariant networks and proposed SE(3)-Transformers. However, since their filters are essentially calculated based on a pair of inputs, the computational complexity is quadratic.

In this work, we further extend the linear equivariant theory to a more general situation, including non-linear cases. Under the framework, we design a new equivariant layer to solve both the spatial-

agnostic problem in convolution-based equivariant models and heavy computation cost problem in most equivariant models.

# 3  A Unified Framework of Previous Group Equivariant Models

In this section, we first briefly review two representative group equivariant models: the linear model G-CNNs [9], and the non-linear model equivariant self-attention [47, 26]. Then, we propose a general framework of previous equivariant models based on the inner relationship among these specific models.

## 3.1  Equivariance

Equivariance indicates that the outputs of a mapping transform in a predictable way with the transformation of the inputs. Formally, a group equivariant map $\Psi$ satisfies that

$$\forall u \in \mathcal{G}, \quad \Psi\left[\mathcal{T}_u[\mathbf{f}]\right] = \mathcal{T}'_u[\Psi[\mathbf{f}]], \tag{1}$$

where $\mathcal{G}$ is a transformation group, $\mathbf{f}$ is an input feature map, and $\mathcal{T}_u$ and $\mathcal{T}'_u$ are group actions, indicating how the transformation $u$ acts on the input and output features, respectively. Besides, since we hope that two transformations $u, v \in \mathcal{G}$ acting on the feature maps successively is equivalent to the composition of transformations $uv \in \mathcal{G}$ acting on the feature maps directly, we require that $\mathcal{T}_u \mathcal{T}_v = \mathcal{T}_{uv}$, where $uv$ is the group product of $u$ and $v$. The same is the case with $\mathcal{T}'_u$.

Now we examine the specific form of the transformation group $\mathcal{G}$. In this work, we focus on the analysis of 2D images defined on $\mathbb{R}^2$. Consequently, we are most interested in the groups of the form $\mathcal{G} = \mathbb{R}^2 \rtimes \mathcal{A}$, resulting from the semi-product ($\rtimes$) between the translation group $\mathbb{R}^2$ and a group $\mathcal{A}$ acts on $\mathbb{R}^2$, e.g., rotations, scalings and mirrorings. This family of groups is referred to as affine groups and their group product rule is:

$$uv = (x_u, a_u)(x_v, a_v) = (x_u + a_u x_v, a_u a_v), \tag{2}$$

where $u = (x_u, a_u)$ and $v = (x_v, a_v)$, in which $x_u, x_v \in \mathbb{R}^2$ and $a_u, a_v \in \mathcal{A}$. For ease of implementation, following [9], we take $\mathcal{A}$ as the cyclic group $\mathcal{C}4$ or the dihedral group $\mathcal{D}4$, then $\mathcal{G}$ becomes $p4$ or $p4m$. As for the group action, we employ the most common regular group action in this work, i.e.,

$$\mathcal{T}_u[\mathbf{f}](v) = \mathbf{f}(u^{-1}v). \tag{3}$$

Here, we only care about the group action over the feature maps defined on $\mathcal{G}$, because we always use a lifting operation to lift the input images defined on $\mathbb{R}^2$ to the feature maps on $\mathcal{G}$, where the equivariance can be preserved properly, as will be shown in Section 3.2.

## 3.2  G-CNNs

Let $\mathbf{f}^{(l)} : \mathcal{X} \to \mathbb{R}^{C_l}$ and $W : \mathcal{G} \to \mathbb{R}^{C_{l+1} \times C_l}$ be the input feature and the convolutional filter in the $l$-th layer, respectively, where $C_l$ denotes the channel number of the $l$-th layer. $\mathcal{X}$ is taken as $\mathbb{R}^2$ for the first layer, and taken as $\mathcal{G}$ for the following layers. Then for any $g \in \mathcal{G}$, the group convolution [29, 7, 1] of $\mathbf{f}^{(l)}$ and $W$ on $\mathcal{G}$ at $g$ is given by

$$\mathbf{f}^{(l+1)}(g) = \Psi[\mathbf{f}^{(l)}](g) = \int_{\mathcal{X}} W(g^{-1}\widetilde{g}) \mathbf{f}^{(l)}(\widetilde{g}) d\mu(\widetilde{g}), \tag{4}$$

where $\mu(\cdot)$ is the Haar measure. When $\mathcal{X}$ is discrete, Eqn. (4) can be rewritten as

$$\mathbf{f}^{(l+1)}(g) = \sum_{\widetilde{g} \in \mathcal{X}} W(g^{-1}\widetilde{g}) \mathbf{f}^{(l)}(\widetilde{g}). \tag{5}$$

G-CNNs essentially generalize the translation equivariance of conventional convolution to a more general group $\mathcal{G}$.

In fact, the first layer maps the 2D images to a function defined on $\mathcal{G}$, while the following layers map one feature map on $\mathcal{G}$ to another. As a result, the computational complexity of the first layer and the following layers are of the order $O(k^2|\mathcal{A}|)$ and $O(k^2|\mathcal{A}|^2)$, respectively, where $k$ is the kernel size in the spatial space. As a result, G-CNNs have a much larger computational cost when $\mathcal{A}$ is large, especially for the intermediate layers. In this work, we employ the first layer of G-CNNs as a lifting operation, and focus on reducing the computation of the latter layers.

## 3.3 Equivariant Attention Networks

Group Equivariant Self-Attention (G-SA) [47, 26] is a representative method of equivariant attention networks, whose form can be simplified as follows:

$$\mathbf{f}^{(l+1)}(g) = \sum_{\widetilde{g} \in \mathcal{G}} \text{Softmax}_{\widetilde{g}} [h_Q^T(\mathbf{f}^{(l)}(g))(h_K(\mathbf{f}^{(l)}(\widetilde{g})) + P_{g^{-1}\widetilde{g}})] h_V(\mathbf{f}^{(l)}(\widetilde{g})), \tag{6}$$

where $h_V : \mathbb{R}^{C_l} \to \mathbb{R}^{C_{l+1}}$, and $h_Q, h_K : \mathbb{R}^{C_l} \to \mathbb{R}^d$ are the embedding functions of values, querys and keys, respectively, which are neural networks in the most general case. $d$ is the dimension of the low dimensional embeddings, and $P_{g^{-1}\widetilde{g}} \in \mathbb{R}^d$ encodes the relative positions of the query $\mathbf{f}^{(l)}(g)$ and the key $\mathbf{f}^{(l)}(\widetilde{g})$.

## 3.4 Generalized Equivariant Framework

As more and more group equivariant structures emerge, researchers start to deduce the most general equivariant structures. To this end, Cohen et al. [7], Kondor et al. [29] and Bekkers [1] proposed a general theory of linear group equivariant structures, which indicates that G-CNNs are the most general equivariant linear layers. Besides, a lot of non-linear equivariant structures appear recently, such as equivariant self-attention layers [47, 26]. This motivates us to investigate a more general framework.

In all, with only slight modification, most of layers in a neural network can be viewed as a kind of aggregation of pair-wise feature interaction as follows:

$$\mathbf{f}^{(l+1)}(g) = \sum_{\widetilde{g} \in \mathcal{G}} H_{g,\widetilde{g}}(\mathbf{f}^{(l)}(g), \mathbf{f}^{(l)}(\widetilde{g})), \tag{7}$$

where the feature aggregation operator $H_{g,\widetilde{g}}(\cdot, \cdot) : \mathbb{R}^{C_l} \times \mathbb{R}^{C_l} \to \mathbb{R}^{C_{l+1}}$ is a mapping indexed by a pair of location $g$ and $\widetilde{g}$, which describes how to aggregate the input feature pair $\mathbf{f}(g)$ and $\mathbf{f}(\widetilde{g})$. In general, the above layer is not equivariant. However, we can find a general constraint for $H_{g,\widetilde{g}}(\mathbf{f}^{(l)}(g), \mathbf{f}^{(l)}(\widetilde{g}))$ to make this layer equivariant over $\mathcal{G}$.

**Theorem 1** *The layer formulated as Eqn.(7) is group equivariant if and only if there is a mapping $\tilde{H}_{\hat{g}} : \mathbb{R}^{C_l} \times \mathbb{R}^{C_l} \to \mathbb{R}^{C_{l+1}}$ which is indexed by a single group element $\hat{g}$, such that, $\forall \mathbf{f}^{(l)}$ and $\forall g, \tilde{g} \in \mathcal{G}$, the layer satisfies:*

$$\sum_{\tilde{g}} H_{g,\tilde{g}}(\mathbf{f}^{(l)}(g), \mathbf{f}^{(l)}(\tilde{g})) = \sum_{\tilde{g}} \tilde{H}_{g^{-1}\tilde{g}}(\mathbf{f}^{(l)}(g), \mathbf{f}^{(l)}(\tilde{g})) \tag{8}$$

**Proof** $\Rightarrow$ *Firstly, $\forall u, g$ and $\widetilde{g} \in \mathcal{G}$,*

$$\mathcal{T}_u \mathbf{f}^{(l+1)}(g) = \mathbf{f}^{(l+1)}(u^{-1}g) = \sum_{\widetilde{g} \in \mathcal{G}} H_{u^{-1}g,\widetilde{g}}(\mathbf{f}^{(l)}(u^{-1}g), \mathbf{f}^{(l)}(\widetilde{g})).$$

*On the other hand,*

$$\sum_{\widetilde{g} \in \mathcal{G}} H_{g,\widetilde{g}}(\mathcal{T}_u \mathbf{f}^{(l)}(g), \mathcal{T}_u \mathbf{f}^{(l)}(\widetilde{g})) = \sum_{\widetilde{g} \in \mathcal{G}} H_{g,\widetilde{g}}(\mathbf{f}^{(l)}(u^{-1}g), \mathbf{f}^{(l)}(u^{-1}\widetilde{g})) = \sum_{\widetilde{g} \in \mathcal{G}} H_{g,u\widetilde{g}}(\mathbf{f}^{(l)}(u^{-1}g), \mathbf{f}^{(l)}(\widetilde{g})).$$

*As $\mathcal{T}_u \mathbf{f}^{(l+1)}(g) = \sum_{\widetilde{g} \in \mathcal{G}} H_{g,\widetilde{g}}(\mathcal{T}_u \mathbf{f}^{(l)}(g), \mathcal{T}_u \mathbf{f}^{(l)}(\widetilde{g}))$,*

$$\Rightarrow \forall \mathbf{f}^{(l)}, g, u, \quad \sum_{\widetilde{g} \in \mathcal{G}} H_{g,u\widetilde{g}}(\mathbf{f}^{(l)}(u^{-1}g), \mathbf{f}^{(l)}(\widetilde{g})) = \sum_{\widetilde{g} \in \mathcal{G}} H_{u^{-1}g,\widetilde{g}}(\mathbf{f}^{(l)}(u^{-1}g), \mathbf{f}^{(l)}(\widetilde{g})).$$

*Let $g \to ug$, we get:*

$$\forall \mathbf{f}^{(l)}, g, u, \quad \sum_{\widetilde{g} \in \mathcal{G}} H_{ug,u\widetilde{g}}(\mathbf{f}^{(l)}(g), \mathbf{f}^{(l)}(\widetilde{g})) = \sum_{\widetilde{g} \in \mathcal{G}} H_{g,\widetilde{g}}(\mathbf{f}^{(l)}(g), \mathbf{f}^{(l)}(\widetilde{g})).$$

*then, we let $u$ to be $g^{-1}$,*

$$\forall \mathbf{f}^{(l)}, g, \quad \sum_{\widetilde{g} \in \mathcal{G}} H_{e, g^{-1}\widetilde{g}}(\mathbf{f}^{(l)}(g), \mathbf{f}^{(l)}(\widetilde{g})) = \sum_{\widetilde{g} \in \mathcal{G}} H_{g, \widetilde{g}}(\mathbf{f}^{(l)}(g), \mathbf{f}^{(l)}(\widetilde{g})).$$

*We denote $\tilde{H}_{g^{-1}\widetilde{g}}(\cdot, \cdot)$ as $H_{e, g^{-1}\widetilde{g}}(\cdot, \cdot)$, we can get exactly the Eqn.(8)*

$\Leftarrow$ *This is obvious.* **Q.E.D**  ∎

From the theorem, we can get a group equivariant layer:

$$\mathbf{f}^{(l+1)}(g) = \sum_{\widetilde{g} \in \mathcal{G}} \widetilde{H}_{g^{-1}\widetilde{g}}(\mathbf{f}^{(l)}(g), \mathbf{f}^{(l)}(\widetilde{g})), \tag{9}$$

which is also the only equivariant form of Eqn. (7). Actually, the above theorem also reveals the essence of equivariance in previous works, i.e., if the relative positions of $(g_1, \widetilde{g_1})$ and $(g_2, \widetilde{g_2})$ are the same, *i.e.,* $g_1^{-1}\widetilde{g_1} = g_2^{-1}\widetilde{g_2} = \hat{g}$, the feature pairs located at the two tuples should be processed equally. In other words, we should employ the same function $\widetilde{H}_{\hat{g}}$ to act on these two input feature pairs.

From this perspective, we can readily see that both the kernel sharing used in G-CNNs, Eqn. (4), and the relative position encoding adopted in the G-SA, Eqn. (6), utilizes the above rule. According to Theorem 1, designing a group equivariant layer becomes much more easily and flexibly than ever, as we only need to design a new function $\widetilde{H}_{\hat{g}}$. In addition, the new formulation provides a more general perspective on the group equivariant layer, i.e., sharing the parameters of function $\widetilde{H}_{\hat{g}}$, which generalizes the kernel sharing schemes in G-CNNs. Based on the above understanding, we can see that if we replace the feature vector in the right hand side of Eqn. (9) with the local patch at group element $g$ and $\widetilde{g}$, respectively, it is still equivariant.

**Proposition 1** *The following layer is equivariant,*

$$\mathbf{f}^{(l+1)}(g) = \sum_{\widetilde{g} \in \mathcal{G}} \widetilde{H}_{g^{-1}\widetilde{g}}(\mathcal{F}_{\mathcal{N}_1(g)}, \mathcal{F}_{\mathcal{N}_2(\widetilde{g})}) \tag{10}$$

*where for $i = 1, 2$, the $\mathcal{F}_{\mathcal{N}_i(g)}$ denote the local patches of $g$, in which $\mathcal{N}_i(g)$ represent $g$'s neighborhood $\{gg' | g' \in \mathcal{N}_i(e)\}$ and $\mathcal{N}_i(e)$ is the predefined neighborhood of the identity element $e \in \mathcal{G}$.*

One remarkable advantages of introducing local patch is that it contains more semantic information than feature vector. Notice, we acquire the local patches by concatenating features in the neighborhoods of $g$ and $\widetilde{g}$ in a predefined order on $\mathcal{N}_1(e)$ and $\mathcal{N}_2(e)$ respectively, *i.e.,* $f(g')$ is concatenated at the same place in $\mathcal{F}_{\mathcal{N}_1(g)}$ as $f(g^{-1}g')$ in $\mathcal{F}_{\mathcal{N}_1(e)}$. We denote the concatenation operator as $\bigcup$, and will discuss the above in detail in Section 4.1, which shows that concatenating features can not only make our framework more flexible, but also help to reduce the computational burden of our newly proposed equivariant layer.

## 4 Efficient Equivariant Layer

A straight-forward and easy case of Eqn. (10) is to adopt $\widetilde{H}_{\hat{g}}, \forall \hat{g} \in \mathcal{G}$, as a multi-layer perceptron (MLP), where the subscript $\hat{g}$ is used to identify different MLPs. However, in Eqn. (10), we have to compute a mapping from two high dimensional vectors to another high dimensional one for each input pair of $g$ and $\widetilde{g}$, which is very expensive. A similar issue exists in computing the attention score in self-attention. To deal with this problem, we decompose $\widetilde{H}$ into the following form to reduce the computation, *i.e.,*

$$\forall \hat{g} \in \mathcal{G}, \quad \widetilde{H}_{\hat{g}}(x, y) = K_{\hat{g}}(x) \odot V(y) \tag{11}$$

where $\odot$ means element-wise product, and $K_{\hat{g}} : \mathbb{R}^{C_l |\mathcal{N}_1(e)|} \rightarrow \mathbb{R}^{C_{l+1}}$ is a kernel generator and $V : \mathbb{R}^{C_l |\mathcal{N}_2(e)|} \rightarrow \mathbb{R}^{C_{l+1}}$ is an encoder. We use $|\cdot|$ to denote the numbers of elements in a set. Hence, we can compute $K_{\hat{g}}(x)$ and $V(y)$ separately. In addition, to further save computation, we split the kernel into several slices along the channels, such that $K_{\hat{g}}$ is shared across these slices,

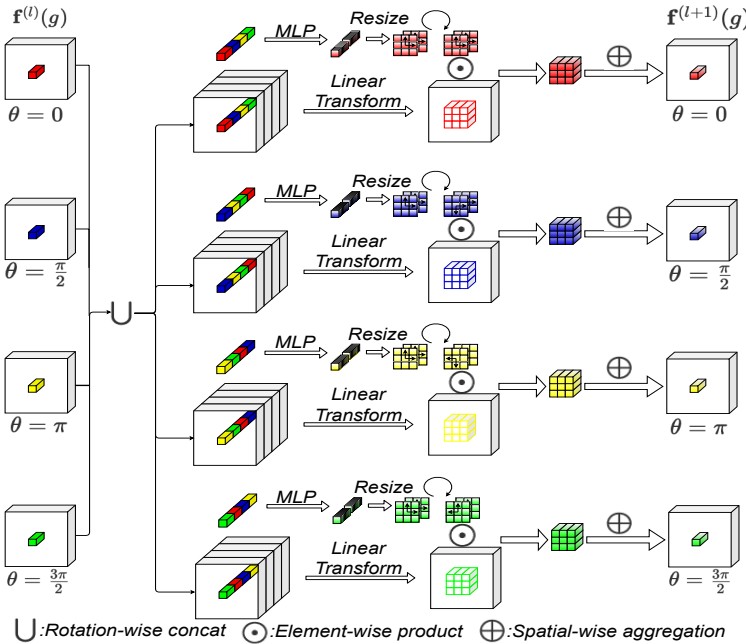

$\bigcup$:Rotation-wise concat   $\odot$:Element-wise product   $\oplus$:Spatial-wise aggregation

Figure 1: An example of our $E^4$-layer on $p4$ group. We firstly concatenate features along rotation dimension in a predefined order on the $\mathcal{C}4$ and then pass them through the MLP and the linear layer to generate kernel and encode features, respectively. After this, the element-wise product is carried out to compute $\widetilde{H}_{g^{-1}\widetilde{g}}$, and finally spatial-wise aggregation is performed to acquire the output. Note that when computing output features at different rotation dimensions, the generated kernel should be rotated by a specific degree to keep the correct relative position.

i.e., $\forall\, 1 \leq i, j \leq C_{l+1}$, $K_{\hat{g}}^i = K_{\hat{g}}^j$ if $i \equiv j \pmod{s}$, where $s$ is the number of slices, and $i$ and $j$ are channel indexes. The $K_{\hat{g}}$ is essentially a dynamic filters which is adaptive to features around $g$, avoiding the spatial-agnostic problem in G-CNNs. Unlike conventional dynamic filters, which are matrices, the output of $K_{\hat{g}}$ is a vector, which can be viewed as a depth wise kernel [5]. This can decouple channel dimension with spatial dimension during feature aggregation to reduce the computational cost. Position information is implicitly encoded in the organized output form of our kernel generator, rather than using explicit positional embedding in the group self-attention layer [26, 47].

In practice, we can view the whole kernel family $\{K_{\hat{g}}\}_{\hat{g}\in\mathcal{G}}$ as the output of a single mapping, i.e., $\widetilde{K}$: $\mathbb{R}^{C_l|\mathcal{N}_1(e)|} \to \mathbb{R}^{|\mathcal{G}|C_{l+1}}$. Then, we resize the output of $\widetilde{K}$ to be a $|\mathcal{G}| \times C_{l+1}$ matrix, with different rows represent different $K_{\hat{g}}$. Namely, if we adopt $\widetilde{K}$ as an MLP, the computations and parameters used for hidden layer are shared across $K_{\hat{g}}$ for different $\hat{g}$, which is another merit of the Eqn. (11). However, there is still a large search space for $\widetilde{H}_{\hat{g}}$, as Eqn (11) is only a special structue of $\widetilde{H}_{\hat{g}}$, we leave a more complete study of $\widetilde{H}_{\hat{g}}$ in the future work.

## 4.1 Implementation on Affine Group

In this section, we design a very efficient equivariant layer based on Eqn. (11) for affine group $\mathbb{R}^2 \rtimes \mathcal{A}$. The computation of the operator is:

$$\mathbf{f}^{(l+1)}(g) = \sum_{\widetilde{g}\in\mathcal{N}(g)} K_{g^{-1}\widetilde{g}} \left( \bigcup_{g'\in\mathcal{N}_1(g)} \mathbf{f}^{(l)}(g') \right) \odot V \left( \bigcup_{\widetilde{g}'\in\mathcal{N}_2(\widetilde{g})} \mathbf{f}^{(l)}(\widetilde{g}') \right). \tag{12}$$

Following the standard practice in computer vision, aggregation is done only on the local neighborhood of $g$, $\mathcal{N}(g)$. To save computation, we choose $\mathcal{N}(g)$ to be only spatial-wise neighborhood, *i.e.,*

$\mathcal{N}(g) = \{g(v, e_{\mathcal{A}}) \mid v \in \Omega\}$, where $\Omega \in \mathbb{R}^2$ and $e_{\mathcal{A}}$ is the identity element of group $\mathcal{A}$. However, aggregating information along spatial neighborhood only discards the information interaction along $\mathcal{A}$, which could lead to a drop in performance [32]. We alleviate the issue by concatenating the feature map along $\mathcal{A}$, i.e., we choose $\mathcal{N}_1(g)$ and $\mathcal{N}_2(g)$ to be $\{g(\mathbf{0}, a) | a \in \mathcal{A}\}$. The order of concatenation is predefined on $\mathcal{A}$. As will be shown in the later experiments, this concatenation does not introduce much computation but can significantly improve performance. Compared to group convolution, such a design enables us to decouple the feature aggregation across the spatial dimension and the $\mathcal{A}$ dimension to further reduce computational cost.

In practice, we adopt the $\widetilde{K}$ as a two layer MLP: $\widetilde{K}(x) = W_2\text{Relu}(W_1 x)$, where $W_1 \in \mathbb{R}^{C_l/r \times C_l|\mathcal{A}|}$, $W_2 \in \mathbb{R}^{|\Omega|s \times C_l/r}$, and $r$ is the reduction ratio which saves both parameters and computation, $s$ is the number of slices defined before. For 2D images, $\Omega$ is usually adopted as a $k \times k$ square mesh grids and $|\Omega| = k^2$, where $k$ is the kernel size. We simply adopt the encoder $V$ as a linear transform: $V(y) = W_3 y$, where $W_3 \in \mathbb{R}^{C_{l+1} \times C_l|\mathcal{A}|}$. For better illustration, we visualize a concrete layer of Eqn. (12) by choosing $\mathcal{G}$ as $p4$ in Figure 1.

## 4.2 Computational Complexity Analysis

In practice, the feature map is defined on discrete mesh grids. We use $h$ and $w$ to denote the height and the width of mesh grids. As the numbers of the input and output channels are usually the same, we assume $C_l = C_{l+1} = c$.

**Parameter Analysis** The number of learnable parameters of $E^4$-layer (12) is $c^2|\mathcal{A}|(1 + 1/r) + csk^2/r$. As $s \ll c$, parameter counts are dominated by the first term when $k$ is not too large, and increasing kernel size will not significantly increase parameter counts, which is shown in later experiments. The parameters count of group convolution layer is $c^2k^2|\mathcal{A}|$. Notice that $(1+1/r) \ll k^2$ and $s/r \ll c|\mathcal{A}|$, parameters count of our $E^4$-layer is significantly less than that of group convolution layer.

**Time Complexity Analysis** The FLOPs of $E^4$-layer and group convolution layer are $(1 + 1/r)c^2|\mathcal{A}|^2hw + (1 + s/r)k^2c|\mathcal{A}|hw$ and $k^2c^2|\mathcal{A}|^2hw$, respectively. Similarly, as $(1 + 1/r) \ll k^2$ and $(1 + s/r) \ll c|\mathcal{A}|$, the FLOPs of $E^4$-layer is significantly lower than that of group convolutional layer.

It can be observed that both the parameter count and FLOPs of our $E^4$-layer are composed of two terms, one depending on $k^2$ and the other not relying on $k$, which is a result of disentangling across spatial dimension with both channels and $\mathcal{A}$ during feature aggregation.

# 5 Experiments

In this section, we conduct extensive experiments to study and demonstrate the performance of our model. The experimental results show that our model has a greater capacity than the group-convolution-based one in terms of parameter efficiency, computational efficiency, data efficiency and accuracy. On the MNIST-rot dataset, we detailedly study the effect of hyperparameters on the number of parameters, computation FLOPs and performance of our model. All the experiments are done on the GeForce RTX 3090 GPU.

## 5.1 Rotated MNIST

The MNIST-rot dataset [33] is the most widely used benchmark to test the equivariant models. It contains 62k $28 \times 28$ randomly rotated gray-scale handwritten digits. Images in the dataset are split into 10k for training, 2k for validation and 50k for testing. Random rotation of digits and only 20 percent of training data of the standard MNIST dataset increases the difficulty of classification.

Table 1: Test error on rot-MNIST(with standard deviation under 5 random seed variations)

| Model | Test error (%) | Params | FLOPs |
|---|---|---|---|
| $p4\_SA$ [47] | 2.54±0.052 | 44.67K | 400M |
| $p4\_CNN$ [9] | 1.79±0.043 | 77.54K | 46.2M |
| $\alpha\_p4\_CNN$[45] | 1.69±0.021 | 73.13K | 27.0M |
| $E^4$-Net (Ours) | **1.29±0.023** | **18.8K** | **17M** |
| $E^4$-Net(Large)(Ours) | **1.17±0.019** | **41.1K** | **36.9M** |

For a fair comparison, we keep both training settings and architectures of our model as close as possible to previous works [9, 47]. In addition, we adopt the $p4$ group to construct all our models in this section. In our first experiment, we adopt our $E^4$-Net given in the supplementary material to make a comparison to previous works. This is a very lightweight model which contains only 18.8K learnable parameters. It is composed of one group convolutional layer which lifts the image to the $p4$ group, six $E^4$-layers and one fully connected layer. Two $2 \times 2$ max-pooling layers are inserted after the first and the third $E^4$-layer to downsample feature maps. The last $E^4$-layer is followed by a global max group pooling layer [9], which takes the maximum response over the entire group, to ensure the predictions invariant to rotations.

Our model is trained using the Adam optimizer [27] for 200 epochs with a batch size of 128. The learning rate is initialized as 0.02 and is reduced by 10 at the 60th, 120th and 160th epochs. The weight decay is set as 0.0001 and no data augmentation is used during training. The results are listed in Table 1. Our models significantly outperform G-CNNs [9] using only about 25% parameters and 40% FLOPs. For G-SA [47], which is a group equivariant stand-alone self-attention model, even performs inferiorly to G-CNNs with much more computational cost. The $\alpha$-p4-CNN model [45] further introduces the attention mechanism to group convolution along both spatial and channel dimensions to enhance the expressiveness of G-CNNs, while our $E^4$-Net still significantly outperforms it with less computational cost. We also experiment with a larger model to further demonstrate the capacity of our model, which is listed in the last line of Table 1.

**Ablation Study of Concatenation**: In the $E^4$-layer (12), we introduce the concatenation operation to enable the disentanglement across the rotation and the spatial information interaction. To study the importance of concatenation, we carry out experiments on the case that neither $K_{\hat{g}}$ nor $V$ in Eqn. (12) use concatenation, i.e., $\mathcal{N}_1(g) = g, \mathcal{N}_2(\widetilde{g}) = \widetilde{g}$. As shown in the first line of Table 2, this leads to a significant drop in performance. This is because if aggregation in Eqn.(12) is done merely in the spatial neighborhoods without concatenation, there is no information interaction along the rotation dimensions. We also experiment the cases using concatenation only in $K_{\hat{g}}$ or $V$, and the performance of both is better than the case without concatenation but is still inferior to the case with concatenation in both $K_{\hat{g}}$ and $V$. This further illustrates the importance of concatenation along $\mathcal{A}$.

Table 2: The effect of concatenation

| Concate | Test error (%) | Params | FLOPs |
|---------|---------------|--------|-------|
| None | 4.10±0.085 | 9.9K | 8.9M |
| only K | 1.96±0.045 | 14.4K | 13M |
| only V | 1.52±0.036 | 14.4K | 13M |
| K&V | **1.29±0.023** | 18.8K | 17M |

**Hyperparameters Analysis**: We investigate the effect of various hyperparameters used in the $E^4$-layer. The reduction ratio $r$ and the slice number $s$ in the $K_{\hat{g}}$ and kernel size $k$ control the computations and parameters of the layer. Based on the baseline model, we vary the three hyperparameters respectively. As shown in the Table 3, improvement is observed when decreasing the reduction ratio and increasing the slice number, with the cost of computational burden increasing. Especially, the improvement of $s = 2$ over $s = 4$ and $r = 1$ over $r = 2$ is marginal, which is attributes to redundancy

Table 3: Hyperparameters Analysis

| Hyperparam | Test error (%) | Params | FLOPs |
|------------|---------------|--------|-------|
| s=1 | 1.45±0.022 | 16.3K | 14.9M |
| s=2 | 1.29±0.023 | 18.8K | 17M |
| s=4 | 1.24±0.026 | 23.9K | 21.2M |
| r=1 | 1.29±0.023 | 18.8K | 17M |
| r=2 | 1.33±0.026 | 13.0K | 12M |
| r=4 | 1.37±0.025 | 10.1K | 9.5M |
| k=3 | 1.46±0.031 | 15.6K | 14.3M |
| k=5 | 1.29±0.023 | 18.8K | 17M |
| k=7 | 1.27±0.021 | 23.8K | 21.1M |

in the kernel [4]. In conclusion, appropriately increasing the reduction ratio $r$ and decreasing the slice number $s$ can help to reduce computational cost while preserving performance. Keeping other hyperparameters fixed, we study the effect of kernel size on our model. In Table 3, the performance peaks when kernel size equals 7. In general, a larger kernel size leads to improved performance due to a larger receptive field. In addition, as explained in Section 4.2, increasing kernel size does not dramatically increase parameters and FLOPs as standard convolution.

## 5.2 Natural Image Classification

In this section, we evaluate the performance of our model on the two common natural image datasets, CIFAR10 and CIFAR100 [30]. The CIFAR-10 and the CIFAR100 datasets consist of $32 \times 32$ images

Table 4: Test error on CIFAR10 and CIFAR100. (with standard deviation under 5 random seed variations)

| Model | CIFAR10 (%) | CIFAR100 (%) | Params | FLOPs |
|---|---|---|---|---|
| R18 | 9.7$\pm$0.43 | 34$\pm$0.76 | 11M | 0.56G |
| $p4$-R18 | 7.53$\pm$0.21 | 27.96$\pm$0.56 | 11M | 2.99G |
| $p4$-$E^4$R18(Ours) | **6.72$\pm$0.14** | **26.59$\pm$0.36** | **5.8M** | **1.85G** |
| $p4m$-R18 | 5.83$\pm$0.17 | 24.95$\pm$0.42 | 10.8M | 5.63G |
| $p4m$-$E^4$R18(Ours) | **4.96$\pm$0.16** | **22.18$\pm$0.46** | **6.0M** | **3.87G** |

belonging to 10 and 100 classes, respectively. Both of the datasets contain 50k training data and 10k testing data. Before training, images are normalized according to the channel means and standard deviations.

In this experiment, we adopt ResNet-18 [22] as the baseline model(short as R18), which is composed of an initial convolution layer, followed by 4 stage Res-Blocks and one final classification layer. Following the standard practice in [9], we replace all the conventional layers with $p4$ ($p4m$) convolutions in R18 and increase the width of each layer by $\sqrt{4}$ ($\sqrt{8}$) to keep the learnable parameters approximately the same. We denote the resulting models as $p4$-R18 ($p4m$-R18). We replace the second group convolution layer in each Res-Block of $p4$-R18 ($p4m$-R18) with our $E^4$-layer, resulting in the $p4$-$E^4$R18 ($p4m$-$E^4$R18). For a fair comparison, all the above models are trained under the same training settings. We use the stochastic gradient descent with an initial learning rate of 0.1, a Nesterov momentum of 0.9 and a weight decay of 0.0005. The learning rate is reduced by 5 at 60th, 120th, and 160th epochs. Models are trained for 200 epochs using 128 batch size. *No data augmentation* is used during training to illustrate data efficiency of our model.

The classification accuracy, parameters count and FLOPs of all models on CIFAR10 and CIFAR100 are reported in Table 4. We can see that models incorporating more symmetry achieve better performance, *i.e.,* R18 $\leq$ $p4$-R18 $\leq$ $p4m$-R18. Our $p4$ and $p4m$ models significantly outperform their counterparts on both CIFAR10 and CIFAR100. Furthermore, our model decreases the parameter count and FLOPs by 45% and 32%, respectively. Notice that the model size reduction is purely caused by the introduction of our $E^4$-layers, as topological connections and width of each layer of $E^4$ model and its counterparts are the same.

**Data Efficiency**: To further study the performance of our model, we train all the models listed in Table 4 on CIFAR10 with different sizes of training data. To be specific, we consider 5 settings, where 1k, 2k, 3k, 4k and 5k training data of each class are randomly sampled from the CIFAR10 training set. Testing is still performed on the original test set of CIFAR10. Other training settings are identical to the above. We visualize the results in Figure 2.

It is observed that the performance gap between $p4$, $p4m$ and $\mathbb{R}^2$ models tend to increase as we reduce the training data. This is mainly because that the prior that the label is invariant to rotations is more important when training data are fewer. The trend is also observed in the gap between our models and their counterparts. For instance, the gap between $p4m$-$E^4$R18 and $p4m$-R18 is 0.87% when training data of each class is 5k, while it is enlarged to 5.22% when training data of each class is reduced to 1k.

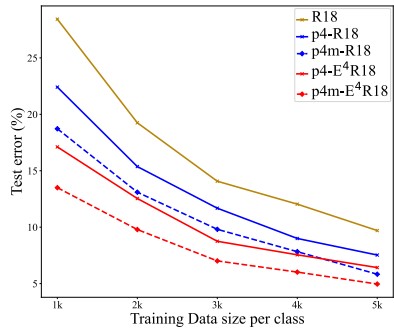

Figure 2: Trend of test error(%) on various training data sizes.

Especially, we observe the line of $p4$-$E^4$R18 intersects with the one of $p4m$-R18, which further indicates that our model is much more data efficient than G-CNNs. As indicated above, symmetry prior is more important when training data are fewer, and the data efficiency of our model implies that $p4$-$E^4$R18 and $p4m$-$E^4$R18 can better exploit the symmetry of data.

# 6   Limitation and Future Work

From the theory perspective, although we extend the general equivaraint framework from linear cases to common non-linear cases, there's two limitations on the generalization: 1) we only focus on layers with such pair-wise interactions proposed in Eqn.(7), and higher-order interactions cases are not included. 2) We only consider regular group action in this framework, which is a special case of general group actions. We leave extending this equivariant framework to these cases as future work.

From the practice perspective, we only give a special implementation of Eqn.(10) in an intuitive insight, and further exploration in the space of equivariant map is in demand. An alternative is to exploit searching algorithms from neural architecture search [42, 38, 69] to find a more powerful and efficient model. Besides this, our $E^4$-layer is slower than G-CNN despite less FLOPs due to convolutions are optimized by many speedup libraries. Our layer is implemented only in a naive way, that is, using the unfold operation followed by a summation operation for the aggregation step. In the future, we will try to implement a customized CUDA kernel for GPU acceleration to reduce training and inference time of our model.

# 7   Conclusions

In this work, we propose a general framework of group equivariant models which delivers a unified understanding on the previous group equivariant models. Based on the new understanding, we propose a novel efficient and powerful group equivariant layer which can serve as a drop-in replacement for convolutional layers. Extensive experiments demonstrate the $E^4$-layer is more powerful, parameter efficient and computational efficient than group convolution layers and their variants. Through a side by side comparison with G-CNNs, we demonstrate our $E^4$-layer can significantly improve data efficiency of equivariant models, which show great potential for reducing the cost of collecting data.

## Acknowledgment

Zhouchen Lin was supported by the NSF China (No.s 61625301 and 61731018), NSFC Tianyuan Fund for Mathematics (No. 12026606) and Project 2020BD006 supported by PKU-Baidu Fund. Yisen Wang is partially supported by the National Natural Science Foundation of China under Grant 62006153, and Project 2020BD006 supported by PKU-Baidu Fund.

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
