# Efficient Equivariant Network
# Supplementary Materials

## A    MNIST-rot Model Architecture

Please refer to Table 5.

Table 5: Architecture of $E^4$-Net on Mnist-rot classification, $p$ means dropout rate.

| Layer | Kernel size | Output channels |
|---|---|---|
| Group convolution | 3×3 | 16 |
| BatchNorm+ReLu | | |
| $E^4$-layer | 5×5 | 16 |
| BatchNorm+ReLu | | |
| Spatial-wise max pooling | 2×2 | |
| $E^4$-layer | 5×5 | 16 |
| BatchNorm+ReLu+Dropout($p$=0.2) | | |
| $E^4$-layer | 5×5 | 16 |
| BatchNorm+ReLu+Dropout($p$=0.2) | | |
| Spatial-wise max pooling | 2×2 | |
| $E^4$-layer | 5×5 | 16 |
| BatchNorm+ReLu+Dropout($p$=0.2) | | |
| $E^4$-layer | 5×5 | 16 |
| BatchNorm+ReLu+Dropout($p$=0.2) | | |
| $E^4$-layer | 5×5 | 16 |
| BatchNorm+ReLu+Dropout($p$=0.2) | | |
| Global max group pooling layer | | |
| Fully connected+Softmax | | 10 |

Here, Global max group pooling [1] is the operation act on each channel:

$$f_i = \max_{g \in \mathcal{G}} f_i(g), \tag{13}$$

and $i$ denotes the channel index. The hyperparameters we use in this architecture are kernel size $k = 5$, reduction ratio $r = 1$, and the number of slices $s = 2$. In the large model, we increase the channel dimension to 24, the number of slices to 12, the reduction ratio to 2, and keep other hyperparameters the same.

## B    Details of CIFAR10 and CIFAR100 Experiments

We take ResNet-18 [2], which is composed of an initial convolution layer, followed by 4 stage Res-Blocks and one final classification layer. Each stage contain 2 Res-Blocks and each

block contain 2 convolution layers. The channel dimensions of each stage of Res-Blocks are $64 \rightarrow 64 \rightarrow 128 \rightarrow 256 \rightarrow 512$. For $p4$-R18 ($p4m$-R18), we replace all the conventional layers with $p4$ ($p4m$) convolutions layers, and modify the channel dimensions at each stage Res-Blocks as $32 \rightarrow 32 \rightarrow 64 \rightarrow 128 \rightarrow 256$ ($22 \rightarrow 22 \rightarrow 44 \rightarrow 88 \rightarrow 176$) to keep the parameter almost invariant. Then, for building our model $p4$-$E^4$R18 ($p4m$-$E^4$R18), the second group convolution layer in each Res-Block of $p4$-R18 ($p4m$-R18) is replaced by our $E^4$-layer with $k = 3$, $r = 2$, and $s = C_l/2$. Here, $C_l$ is the channel dimension at that layer.

## C    Proof of Theorem

**Proposition 1** *The following layer is equivariant, where for $i = 1, 2$, $\mathcal{N}_i(e)$ is the neighbor of the group identity element $e \in \mathcal{G}$ and $\mathcal{N}_i(g) := \{gg'|g' \in \mathcal{N}(e)\}$ is the neighbor around the group element g.*

$$\mathbf{f}^{(l+1)}(g) = \sum_{\widetilde{g} \in \mathcal{G}} \widetilde{H}_{g^{-1}\widetilde{g}}(\mathcal{F}_{\mathcal{N}_1(g)}, \mathcal{F}_{\mathcal{N}_2(\widetilde{g})}) \tag{14}$$

**Proof 1** $\forall u, g \in \mathcal{G}$,

$$\sum_{\widetilde{g} \in \mathcal{G}} \widetilde{H}_{g^{-1}\widetilde{g}}(\mathcal{T}_u \mathcal{F}_{\mathcal{N}_1(g)}, \mathcal{T}_u \mathcal{F}_{\mathcal{N}_2(\widetilde{g})}) \tag{15}$$

$$= \sum_{\widetilde{g} \in \mathcal{G}} \widetilde{H}_{g^{-1}\widetilde{g}}\left(\left(\bigcup_{g' \in \mathcal{N}_1(g)} \mathcal{T}_u \mathbf{f}^{(l)}(g')\right), \left(\bigcup_{\widetilde{g}' \in \mathcal{N}_2(\widetilde{g})} \mathcal{T}_u \mathbf{f}^{(l)}(\widetilde{g}')\right)\right) \tag{16}$$

$$= \sum_{\widetilde{g} \in \mathcal{G}} \widetilde{H}_{g^{-1}\widetilde{g}}\left(\left(\bigcup_{g' \in \mathcal{N}_1(g)} \mathbf{f}^{(l)}(u^{-1}g')\right), \left(\bigcup_{\widetilde{g}' \in \mathcal{N}_2(\widetilde{g})} \mathbf{f}^{(l)}(u^{-1}\widetilde{g}')\right)\right) \tag{17}$$

$$= \sum_{\widetilde{g} \in \mathcal{G}} \widetilde{H}_{g^{-1}\widetilde{g}}\left(\left(\bigcup_{g'' \in \mathcal{N}_1(u^{-1}g)} \mathbf{f}^{(l)}(g'')\right), \left(\bigcup_{\widetilde{g}'' \in \mathcal{N}_2(u^{-1}\widetilde{g})} \mathbf{f}^{(l)}(\widetilde{g}'')\right)\right) \tag{18}$$

$$= \sum_{\overline{g} \in \mathcal{G}} \widetilde{H}_{g^{-1}u\overline{g}}\left(\left(\bigcup_{g'' \in \mathcal{N}_1(u^{-1}g)} \mathbf{f}^{(l)}(g'')\right), \left(\bigcup_{\widetilde{g}'' \in \mathcal{N}_2(\overline{g})} \mathbf{f}^{(l)}(\widetilde{g}')\right)\right) \tag{19}$$

$$= \sum_{\overline{g} \in \mathcal{G}} \widetilde{H}_{(u^{-1}g)^{-1}\overline{g}}\left(\mathcal{F}_{\mathcal{N}_1(u^{-1}g)}, \mathcal{F}_{\mathcal{N}_2(\overline{g})}\right) \tag{20}$$

$$= \mathbf{f}^{(l+1)}(u^{-1}g) = \mathcal{T}_u \mathbf{f}^{(l+1)}(g) \tag{21}$$

*Line (17) to line (18) is by substitution $g'' \to u^{-1}g'$ and $\widetilde{g}'' \to u^{-1}\widetilde{g}''$, in addition to the definition of neighbor of each group element which satisfy $u^{-1}\mathcal{N}_i(g) = \mathcal{N}_i(u^{-1}g)$ for $i = 1, 2$. Line (15) equals line (21) indicates equivariance of layer (14).* **Q.E.D** ∎

## D    Result on ImageNet

We conducte the experiments on ImageNet to demonstrate the performance of our model. We choose R18, p4-R18 and p4-R18 which are described in section 5.2, except that the last fully connected layer are replaced to deal with classification of 1000 category. In the experiments, we adopt commonly used data augmentation as in [2] and train all these models for 120 epochs utilizing the Stochastic Gradient Descent (SGD) optimizer with the momentum of 0.9 and the weight decay of 0.0001. The learning rate initiates from 0.3 and gradually approaches zero following a half-cosine function shaped schedule. The results are listed in Table 6. Our model significantly outperforms G-CNNs with smaller model size on the ImageNet which is consistent with results on CIFAR.

## E    Two Special Case

In this section, we explicit show how to construct concrete $\tilde{H}_g$ to get G-Conv and Attentive G-Conv. 1) For G-Conv, by requiring $\tilde{H}_{g^{-1}\tilde{g}}(f^{(l)}(g), f^{(l)}(\tilde{g})) = W_{g^{-1}\tilde{g}}f^{(l)}(\tilde{g})$, Eqn.(7) reduce to G-Conv.

Table 6: Results on ImageNet.

| Model | Top1 error (%) | Top5 error (%) |
|---|---|---|
| R18 | 28.71 | 9.80 |
| $p4$R18 [1] | 25.30 | 7.67 |
| $E^4$-Net (Ours) | **23.82** | **6.91** |

2)The group equivariant attentive convolutional layer introduced in [3] can be written as following forms:

$$f^{(l+1)}(g) = \sum_{\tilde{g} \in \mathcal{G}} \mathbf{A}[f^{(l)}]_{g,\tilde{g}} W_{g^{-1}\tilde{g}} f^{(l)}(\tilde{g}) \tag{22}$$

where the attention weight have to satisfy the following constraint:

$$\forall f^{(l)}, g, \bar{g}, \tilde{g}, \quad \mathbf{A}[\mathcal{T}_{\bar{g}} f^{(l)}]_{g,\tilde{g}} = \mathbf{A}[f^{(l)}]_{\bar{g}^{-1}g, \bar{g}^{-1}\tilde{g}} \tag{23}$$

In the Eqn.(7), we adopt neighborhood $\mathcal{N}_1(\cdot)$ to be the whole group, *i.e.*, $\mathcal{N}_1(\cdot) = \mathcal{G}$, and $\mathcal{N}_2(g)$ to contain only $g$ itself. $\tilde{H}_{\hat{g}}$ is chosen to be:

$$\tilde{H}_{\hat{g}}(x, y) = \tilde{A}_{\hat{g}}[x] W_{\hat{g}} y. \tag{24}$$

So the layer Eqn.(7) is reduced to:

$$f^{(l+1)}(g) = \sum_{\tilde{g} \in \mathcal{G}} \tilde{H}_{g^{-1}\tilde{g}} \left( \mathcal{F}_{\mathcal{N}_1(g)}, \mathcal{F}_{\mathcal{N}_2(\tilde{g})} \right) = \sum_{\tilde{g} \in \mathcal{G}} \tilde{\mathbf{A}}_{g^{-1}\tilde{g}} \left[ \bigcup_{g' \in \mathcal{N}_1} f^{(l)}(g') \right] W_{g^{-1}\tilde{g}} f^{(l)}(\tilde{g}) \tag{25}$$

Next, we will prove that Eqn.(22), under the condition of Eqn.(23), is a special case of Eqn.(25).

Note that $f^{(l)}$ represent the whole feature map, which can be written as the concatenation of all feature vector $f^{(l)}(g)$ in a predefined order: $\bigcup_{g \in \mathcal{N}_1(e)} f^{(l)}(g)$ where $\mathcal{N}_1(e) = \mathcal{G}$. Take it into Eqn.(23), we can get

$$\forall g, g', \bar{g}, \quad \mathbf{A} \left[ \mathcal{T}_{\bar{g}} \left( \bigcup_{g' \in \mathcal{N}_1(e)} f^{(l)}(g') \right) \right]_{g,\tilde{g}} = \mathbf{A} \left[ \bigcup_{g' \in \mathcal{N}_1(e)} f^{(l)}(g') \right]_{\bar{g}^{-1}g, \bar{g}^{-1}\tilde{g}} \tag{26}$$

For Eqn.(26), let $\bar{g}^{-1}g \to g_1$, and $\bar{g}^{-1}\tilde{g} \to g_2$, we can find:

$$\mathbf{A} \left[ \mathcal{T}_{\bar{g}} \left( \bigcup_{g' \in \mathcal{N}_1(e)} f^{(l)}(g') \right) \right]_{\bar{g}g_1, \bar{g}g_2} = \mathbf{A} \left[ \bigcup_{g' \in \mathcal{N}_1(e)} f^{(l)}(g') \right]_{g_1, g_2} \tag{27}$$

According to the definition of group action and neighbor,

$$\mathcal{T}_{\bar{g}} \left( \bigcup_{g' \in \mathcal{N}_1(e)} f^{(l)}(g') \right) = \bigcup_{g' \in \mathcal{N}_1(\bar{g}^{-1})} f^{(l)}(g') \tag{28}$$

Take Eqn.(28) into Eqn.(27) and let $\bar{g} = g_1^{-1}$, we have:

$$\mathbf{A} \left[ \bigcup_{g' \in \mathcal{N}_1(g_1)} f^{(l)}(g') \right]_{e, g_1^{-1}g_2} = \mathbf{A} \left[ \bigcup_{g' \in \mathcal{N}_1(e)} f^{(l)}(g') \right]_{g_1, g_2} = \mathbf{A} \left[ f^{(l)} \right]_{g_1, g_2} \tag{29}$$

If we let $\tilde{\mathbf{A}}_{\hat{g}}[\cdot] = \mathbf{A}[\cdot]_{e,\hat{g}}$, then Eqn.(25) become Eqn.(22) by substitution of Eqn.(29). This indicates attentive G-Conv can be seen as the special cases of the Eqn.(25). This demonstrate the attentive G-Conv can be viewed as a special case of our framework.