# OpenReview forum: "Efficient Equivariant Network"
_NeurIPS.cc/2021/Conference — NeurIPS 2021 Poster_

### Official Review · Reviewer_3YmE · 2021-07-16

**Rating:** 7
**Confidence:** 5

**Summary:**

The paper describes a generalization of group equivariant neural networks that typically rely on linear operators such as the group convolution, or pseudo linear operators such as attentive group convolution. The authors take a look at the aggregation function of the convolution operator (which is usually linear via a kernel times feature values followed by sum aggregation) and replaces it with a non-linear function that generally could depend not just on relative positions, but also on the central and neighbor feature values. The result is a more non-linear operator than group convolutions, which are a special case of this framework.

The paper is accurate and has an appropriate experimental section with decent ablation studies. The proposed work compares favorably compared to the baselines both in terms of accuracy, parameters and nr of operations (FLOPs).

**Ethical Concerns:**

None.

**Limitations And Societal Impact:**

The ablation study is done on MNIST and conclusion may not generalize to more challenging datasets. Otherwise I see no concerns.

**Main Review:**

I think the paper describes a great idea of modifying the main workhorse of G-CNNs (the group convolution) to something non-linear. Also the paper focusses on efficiency which is a relevant aspect of G-CNNs. The paper performs a decent set of abblation studies, though these are done on MNIST, so it may not generalize to more challenging datasets, but the method is further validated on CIFAR10 and CIFAR100. Since the paper focussed on efficiency it was a pity not to see the method being applied to problems that actually require efficient implementations (such as e.g. imagenet). I think this is one of the main limitations of the paper, but up to some other concerns (see below), I think it is a great paper.

I think he paper could mainly improve by presenting additional intuition behind the proposed method and discuss the main mechanism that motivates why it should work better (see second upcoming concern). This could possibly be achieved by explicitly framing conventional methods as special cases of this framework, which enables to precisely point to differences that could explain why the current method should work better. I find this important particularly because the method is presented as a generalization of other works, and it would be great to explicitly see related works  as special cases. I think this would help interpretation.

All in all I have the following comments:

[Thm 1 + proof | concern about input dependency on g and \tilde{g}]
I think the construction is not very transparent with respect to the dependency of of H to it’s inputs. Namely in equation 8, the inputs of H_{g,\tilde{g}} are denoted as arbitrary open “slots” (with \cdot) for arbitrary inputs. However, these inputs in practice do depend again on g and \tilde{g}, as given in equation 7. With this in mind I don't think the proof is formally correct and is in my opinion even misleading. Though when explicitly taking the dependency into account the theorem and proof is valid . After 154 I suggest not writing \cdot for the inputs but f(g) and f(\tilde{g}) respectively. (Or please correct me if my concern is ill founded).

Related to this concern: In equation 11 x and y still depend on g or \tilde{g}. This dependency seems important but is currently obscured, is this a problem?

[related work and interpretation of the generalization]
Then, the proof of thm. 1 follows the exact same structure as thm.1 of [Bekkers] and reference [23] for the attentive version. It would be great if these works and other related works could be discussed in more detail when it comes to claims regarding this work generalizing many group equivariant works. Just to be clear, I do not dispute this claim, it is very well founded, it is just that the paper contains sentence such as “this work… delivers a unified understanding on the previous group equivariant models” (line 357). In my opinion, the understanding of this unified view could be better delivered as the equation is quite abstract and I think can be better understood with some specific (mathematically precises) special cases such as g-conv or attentive g-conv.

For example, an additional intuitive explanation of equation 12 would be very helpful. Why is this layer more powerful than standard convs?

ref:
[Bekkers] Erik J Bekkers. B-spline cnns on lie groups. In International Conference on Learning Representations, 2019.

[solution to the spatial agnostic problem?]
Then I have doubts regarding the claim of solving “the spatial-agnostic problem”. I do not understand how this problem is solved. By construction equivariant methods have to be spatial agnostic otherwise they can’t be equivariant and thus operations only depend on relative positions/transformations. The only difference now is that the convolution kernels are conditioned on the feature values at the central or neighbor locations. Maybe it is mentioned somewhere in other words, but do I understand correctly that merely the dependency of the conv kernel (as in deformable convs), or aggregation function on local features solves the spatial agnostic problem? My apologies for possibly misinterpreting parts of the paper, but perhaps the notions of equivariance vs “conditioning” can be decoupled and explicitly discussed in terms of the spatial agnostic problem. For me this part was slightly confusing.

[Practical question]
How is the aggregation function/convolution kernel parametrized. In continuous kernel convolutions such as e.g. PointConvs [Wu et al.] or LieConvs [Finzi et al.], the kernel is parametrized by an MLP (though it does not depend on neighborhood feature values). Then for every possible g^{-1}\tilde{g} the kernel is defined. Is this also the case in this paper, or is the kernel (e.g. in eq 12) indexed for a finite set of possible values for g^{-1}\tilde{g}? (made possible because only a discrete group is considered)

refs
[Wu et al.] Wenxuan Wu, Zhongang Qi, and Li Fuxin. Pointconv: Deep convolutional networks on 3d point clouds. In Proceedings of the IEEE/CVF Conference on Computer Vision and Pattern Recognition, pages 9621–9630, 2019.
[Finzi et al.] Marc Finzi, Samuel Stanton, Pavel Izmailov, and Andrew Gordon Wilson. Generalizing convolutional neural networks for equivariance to lie groups on arbitrary continuous data. In International Conference on Machine Learning, pages 3165–3176. PMLR, 2020.

[Minor comments. ]
When equation 10 is presented I didn’t understand why this property was important. But then later on this is used for efficient implementations where the sub-group part is handled in one step by aggregating neighborhood information. Perhaps the section about eq 10 could be better introduced with such a motivation.

Line 217. What is “s”. I think this symbol is not defined.
Lin 292. Same here, I don’t understand what a “slice numer s” is.


Regarding implementation of eq 12. The output of V represents transformed feature values of some neighboorhood, and it should follow the same ordering as the kernel values outputted by K. Is this automatically obtained or does one have to practically code this?

Line 320, why is only the second group conv layerin a res-block replaced and not all?

Section 5.2. Since the focus is on efficiency I was a bit dissapointed not to see any application to large datasets that actually require efficiency. Where there still practical limitations?

The FLOP analysis is great, still I would like to get a sense of computation times. For example, I can imagine that all the stacking of features and what not leads to quite some overhead which, despite less FLOPs, doesn’t lead to a reduction in computation time.





**Time Spent Reviewing:**

8

---

> ### Author Response · Authors · 2021-08-10
> **Reply to reviewer 3YmE(1)**
>
> Thank you for your appreciation of our idea and the constructive suggestions for our paper. We will revise our paper carefully according to your comments. Here are our answers to your comments and questions.
>
> Q1: [Thm 1 + proof | concern about input dependency on g and $\tilde{g}$] I think the construction is not very transparent with respect to the dependency of H to it’s inputs. Namely in equation 8, the inputs of $H_{g,\tilde{g}}$ are denoted as arbitrary open “slots” (with $\cdot$) for arbitrary inputs. However, these inputs in practice do depend again on g and $\tilde{g}$, as given in equation 7. With this in mind I don't think the proof is formally correct and is in my opinion even misleading. Though when explicitly taking the dependency into account the theorem and proof is valid . After 154 I suggest not writing $\cdot$ for the inputs but $f(g)$ and $f(\tilde{g})$ respectively. (Or please correct me if my concern is ill founded). Related to this concern: In equation 11 x and y still depend on g or $\tilde{g}$. This dependency seems important but is currently obscured, is this a problem?
>
> A1: We agree that the description and proof of theorem1 are a little bit misleading. To deal with your concern, we update them in the following anonymous link, theorem.pdf, and it would not affect the conclusion, Eqn.9 is the only equivariant form of Eqn.7, of our paper.
> In Eqn.11, we don't think it is a problem. In this equation, we just want to emphasize the function form of $\tilde{H}$, so we take $\hat{g}$ as index and $x,y$ to refer to its inputs. As for showing the layer form, in Eqn.12, $f(g)$ and $f(\tilde{g})$ should be concerned.
>
> https://www.dropbox.com/sh/nxobxarp9ond1i7/AAAv5H5X75CazIVSrH9uWFx_a?dl=0
>
> Q2: [related work and interpretation of the generalization] Then, the proof of thm. 1 follows the exact same structure as thm.1 of [Bekkers] and reference [23] for the attentive version. It would be great if these works and other related works could be discussed in more detail when it comes to claims regarding this work generalizing many group equivariant works. Just to be clear, I do not dispute this claim, it is very well founded, it is just that the paper contains sentence such as “this work… delivers a unified understanding on the previous group equivariant models”(line 357). In my opinion, the understanding of this unified view could be better delivered as the equation is quite abstract and I think can be better understood with some specific (mathematically precises) special cases such as gconv or attentive g-conv. For example, an additional intuitive explanation of equation 12 would be very helpful. Why is this layer more powerful than standard convs?
>
> A2: [Bekker] gives a framework on the general linear equivariant map between feature map defined on two homogeneous spaces, hence the group convolution can be viewed as a special case of the framework. Compared to it, our work is a generalization of group equivariant network in another direction, which aims to include both the linear g-conv layer and the non-linear g-sa layer. The paper [23] introduced an attention weight on the top of g-conv, and their thm1 gives a sufficient and necessary condition for the attention weight to make the layer equivariant, but does not provide a general solution for the attention weight to satisfy the condition in the thm, which increases the difficulty to design such a layer. As shown below, this layer can be formalized as a special case of our Eqn.10 with slight modification. Moreover, we can prove that the special case is the only solution of the attentive g-conv, and this indicates that our framework is more general than attentive g-conv.  We upload the proof to the following anonymous link, special case.pdf
>
> https://www.dropbox.com/sh/nxobxarp9ond1i7/AAAv5H5X75CazIVSrH9uWFx_a?dl=0
>
> We show reasons why the layer we proposed in Eqn.12 is more powerful than the standard G-CNN as follows:
> 1) The kernel is dependent on the features, so it is position-specific. It avoids the spatial-agnostic problem which would be stated in the following Q&A.
> 2) As lines 213-215 say, the layer decouples the feature aggregation across the spatial dimension and extra geometric dimension. For the standard G-CNN, the information at their spatial dimension and extra geometric dimension is closely mixed in processing, leading to heavy redundancy that would degrade the generalization ability of the network. In our work, the spatial-alone, extra-geometric-alone and spatial-extra-geometric information independently propagates through the network, which is more efficient than the standard G-CNN.
>
> Q3: [solution to the spatial agnostic problem?] Then I have doubts regarding the claim of solving “the spatial-agnostic problem”. I do not understand how this problem is solved. By construction equivariant methods have to be spatial agnostic otherwise they can’t be equivariant and thus operations only depend on relative positions/transformations. The only difference now is that the convolution kernels are conditioned on the feature values at the central or neighbor locations. Maybe it is mentioned somewhere in other words, but do I understand correctly that merely the dependency of the conv kernel (as in deformable convs), or aggregation function on local features solves the spatial agnostic problem? My apologies for possibly misinterpreting parts of the paper, but perhaps the notions of equivariance vs “conditioning” can be decoupled and explicitly discussed in terms of the spatial agnostic problem. For me this part was slightly confusing.
>
> A3: We first give a brief introduction to the spatial-agnostic problems. As we mentioned in lines 30-32, the spatial-agnostic problem ([Su et al.(2019)] and [Wu et al(2018)]) is that the kernel sharing scheme lacks the ability to adapt kernels to diverse feature patterns with respect to different spatial positions. To speak specifically, consider the gradient of a convolution layer during the training process, different descend directions are applied to minimize the loss at each position. If the kernel is spatially shared, the loss gradients at each position will be globally pooled to train the kernel, leading to the sub-optimal kernel learning that the global gradient could be zero while the local gradient is non-zero. Equivariant property has no correlation to the spatial-agnostic problem, but one of their common architecture, G-CNN, suffers from this problem. Here, for G-CNN, the ‘spatial’ is not in a narrow sense in the 2-dimension plane, it refers to the group space. Although bringing in more kernels can alleviate the problem to some extent, it largely increases the number of parameters and memory footprints and is inefficient, especially for G-CNN.
>
> One of the solutions is to make the kernel position-specific, as discussed in [20,36,37], which can alleviate the unshareable descend direction issue and take advantage of gradients at each position.
> Our implementation of $E^4$layer avoids the spatial-agnostic issue by designing a kernel generating function $K_{\hat{g}}(\cdot)$ that imposes the kernels to condition on features and its neighbor features at each group position. Then, the position-specific kernels can take advantage of gradients at each position, making the network more efficient and flexible.
>
> ref: [Su et al.(2019)] Su, Hang, et al. "Pixel-adaptive convolutional neural networks." IEEE/CVF Conference on Computer Vision and Pattern Recognition. 2019.
>
> [Wu et al(2018)] Wu, Jialin, et al. "Dynamic filtering with large sampling field for convnets." European Conference on Computer Vision (ECCV). 2018.
>
> Q4: [Practical question] How is the aggregation function/convolution kernel parametrized. In continuous kernel convolutions such as e.g. PointConvs [Wu et al.] or LieConvs [Finzi et al.], the kernel is parametrized by an MLP (though it does not depend on neighborhood feature values). Then for every possible $g^{-1}\tilde{g}$ the kernel is defined. Is this also the case in this paper, or is the kernel (e.g. in eq 12) indexed for a finite set of possible values for $g^{-1}\tilde{g}$? (made possible because only a discrete group is considered)
>
> A4: In this paper, it is exactly the latter case that the $K_{g^{-1}\tilde{g}}$ in Eqn.12 is indexed for a finite set of possible values for $g^{-1}\tilde{g}$ and we share the computations of $K$ for different index values $g^{-1}\tilde{g}$ to save computations, which is introduced in detail in lines 197-202. Of course, for continuous kernels, we can take a similar method like PointConvs and LieConvs to parameterize it with an MLP.
>
> Q5: [Minor comments. ] When equation 10 is presented I didn’t understand why this property was important. But then later on this is used for efficient implementations where the sub-group part is handled in one step by aggregating neighborhood information. Perhaps the section about eq 10 could be better introduced with such a motivation.
>
> A5: We’ll propose a more motivational introduction for Eqn.10 in the revision.
>
> Q6: Line 217. What is “s”. I think this symbol is not defined. Lin 292. Same here, I don’t understand what a “slice numer s” is.
>
> A6: The slice number $s$ is defined in lines 188-189. Maybe it is not obvious, so we’ll emphasize it in the revision.
>
> Q7: Regarding implementation of Eqn.12. The output of V represents transformed feature values of some neighboorhood, and it should follow the same ordering as the kernel values outputted by K. Is this automatically obtained or does one have to practically code this?
>
> A7: We have to practically code this ordering, which is actually a permutation of the indices.

---

> > ### Author Response · Authors · 2021-08-10
> > **Reply to reviewer 3YmE(2)**
> >
> > Q8: Line 320, why is only the second group conv layerin a res-block replaced and not all?
> >
> > A8: We have tried to replace both group conv layers in the res-block of p4-R18 with our $E^4$-layer. The classification error is 8.66\% on the CIFAR10 dataset under the same training setting described in section 5.2, with the parameters 2.76M and flops 0.69G. We conjecture that the degraded performance compared to p4-R18 is due to too small model size, 2.76M vs 11M. So to make a trade-off of accuracy and model size, we only replace the second group conv layer in a res-block.
> >
> > Q9: Section 5.2. Since the focus is on efficiency I was a bit dissapointed not to see any application to large datasets that actually require efficiency. Where there still practical limitations?
> >
> > A9: We have conducted the experiments on ImageNet to demonstrate the performance of our model. We choose R18, p4-R18 and p4-$E^4$R18 which are described in section 5.2, except that the last fully connected layer are replaced to deal with classification of 1000 categories. In the experiments, we adopt commonly used data augmentation as in [10] and train all these models for 120 epochs utilizing the Stochastic Gradient Descent (SGD) optimizer with the momentum of 0.9 and the weight decay of 0.0001. The learning rate initiates from 0.3 and gradually approaches zero following a half-cosine function shaped schedule. No training tricks are adopted, and the results are listed in Table V. Our model significantly outperforms G-CNNs with smaller model size on the ImageNet which is consistent with results on CIFAR.
> >
> > Table V: Results on ImageNet.
> >
> >  | Model | top1 error (%) |top5 error (%) |
> >  | --- | --- | --- |
> >  | R18 | 28.71 | 9.8 |
> >  | p4-R18 | 25.3 | 7.67|
> >  | p4-$E^4$R18 | 23.8 |6.9 |
> >
> > Q10:The FLOP analysis is great, still I would like to get a sense of computation times. For example, I can imagine that all the stacking of features and what not leads to quite some overhead which, despite less FLOPs, doesn’t lead to a reduction in computation time.
> >
> > A10: The last two lines of Table VI give the results of inference time of p4-R18 and p4-$E^4R18$ for a single image on One GPU (RTX 3090). Our $E^4$-layer is slightly slower than the G-conv layer despite lower flops. This is due to that convolutions are optimized by many speedup libraries, while our $E^4$layer is implemented only in a naive way, that is, using the unfold operation followed by a summation operation for the aggregation step. The unfold operation actually severely slows down the inference of $E^4$ layer. For a better illustration, we also implement the G-conv with an unfolding operator and report the results in the first two lines of Table VI (as the g-conv also exists in resblock of p4-$E^4R18$, whose speed is also affected), we can see that p4-$E^4R18$ is faster than p4-R18 model in this setting. Hence, a well-optimized implementation is very important for speeding up a layer. A customized CUDA kernel implementation for GPU acceleration is expected for $E^4$-layer to approach its theoretical speedup over the convolution layer, but this is beyond this research and we leave this as the future work.
> >
> > Table VI: Inference time analysis
> >
> > | Model |Inference time (s) |
> > | --- | --- |
> > | p4_R18 (unfold) | 0.11 |
> > | p4_$E^4$R18 (unfold) | 0.079 |
> > | p4_R18 | 0.026 |
> > | p4_$E^4$R18 | 0.032 |

---

> > > ### Comment · Reviewer_3YmE · 2021-08-20
> > > **Thank you for the improvements.**
> > >
> > > Thank you for the elaborate answer and shown points of improvement! I'm glad to see the following points of improvement:
> > >
> > > * The notationally issues have been resolved as shown in the anonymous pdf files. The original cdot notation in Thm 1 suggests that in the aggregation functions (that are indexed by $g,\tilde{g}$) the group elements $g,\tilde{g}$ have nothing to do with the input, and can be any vector input. But this is not the case, the input should be feature maps sampled at those locations $g$ and $\tilde{g}$. I think in derivations it is OK to make such notations but in a theorem that should be interpretable as a stand-aline statement. I am happy to see that this is now done. Thank you!
> > > * I'm also happy to see that related work is explicitly written as a special case of this work. Considering this, perhaps a table with different special cases of $\tilde{H}_{g^{-1}\tilde{g}}(F_n(g),F_n(\tilde{g}))$ would also be a helpful addition, though not necessary.
> > > * Thank you for the timings and explanation thereof.
> > > * Thank you for the additional experiments.
> > > I think all the above certainly improves the paper.
> > >
> > > I finally agree with the comment of reviewer F1d8 which is a matter of presentation of the generality of the method. It would improve the paper if the type of generalization could be better framed. E.g. explicitly state in what sense the method is more general (convolutions apply linear transformations to the features whereas the proposed trafos/aggregation is non-linear; the feature transformations are based on pair-wise interactions, but as reviewer F1d8 pointed out, this could even be further generalized; ...).

---

> > > > ### Author Response · Authors · 2021-08-20
> > > > **Thank you for your feedback.**
> > > >
> > > > Thank you very much for your feedback, and we are happy to see your recognition of our reply. Here is our reply for your additional concern:
> > > >
> > > > Since the most common and successful equivariant layers such as g-conv and g-sa can be viewed as pair-wise interactions (Eqn.7), our method mainly focuses on the most general form of equivariant layer of such pair-wise interactions, which is presented in thm1. From this perspective, our framework provides a tool for designing new equivariant layers of such form. We agree that our framework does not provide the most general form of non-linear equivariant layer, as it does not includes higher-order interactions. We'll make the claim and meaning of the generalization clearer in the revision. On the other hand, as higher-order interactions may bring in more computation costs, we leave their general equivariant form and corresponding efficient equivariant implementation as future work.
> > > >
> > > > Best regards, The Authors

---

### Official Review · Reviewer_y9u4 · 2021-07-16

**Rating:** 7
**Confidence:** 4

**Summary:**

This paper presents a general framework for group equivariant models which generalizes group equivariant CNNs (G-CNNs) and equivariant self-attention layers. For practical use, the paper proposes a particular framework, called E4-Net, which can significant improve previous G-CNNs with smaller model size.

**Limitations And Societal Impact:**

yes

**Main Review:**

Two main contributions of the paper are as follows:
- The authors propose a general framework of group equivariant models which generalizes G-CNNs and equivariant self-attention layers. The authors suggest a layer $\psi:\mathbf{f}^l \mapsto \mathbf{f}^{l+1}$ where $\mathbf{f}^{l+1}(g)$ is defined to be a sum over $\tilde{g} \in \mathcal{G}$ of values of a function, say $H$, at $g,\tilde{g},\mathbf{f}^l(g)$ and $\mathbf{f}^l(\tilde{g})$. It is proved in Theorem 1 of the paper that $\psi$ is equivariant if and only if $H$ can be viewed as a function at $g^{-1}\tilde{g},\mathbf{f}^l(g)$ and $\mathbf{f}^l(\tilde{g})$.
- A special case for the function $H$ is used to construct E4-Net, a new group equivariant layers. In E4-Net, the function $H$ is considered to be a point-wise product of a kernel generator $K_{g^{-1}\tilde{g}}(\mathbf{f}^l(g))$ and an encoder $V(\mathbf{f}^l(\tilde{g}))$. Experiments on Rotated MNIST and CIFAR shows that E4-Net outperforms classical G-CNNs with less numer of parameters and time complexity.

The results are novel and interesting. The paper is well-written. Besides, I have some comments as follows:

1. The construction for the kernel generator $K$ is described in detail in page 5. But I do not see any clear construction for the encoder $V$. It seems to me from Figure 1 that the encoder $V$ is chosen to be a linear map. If yes, is there any reason for choosing $V$ to be a linear map rather than an MLP?

2. An explanation for the relation of E4-layer with G-Conv layer and G-SA layer is needed.

3. In experiments, comparisons of the accuracy of E4-Net with G-CNNs are reasonable. Why do not you compare with G-SA neural nets?

Minor comments:
- page 4, line 132: "querys" --> "queries"

**Time Spent Reviewing:**

16

---

> ### Author Response · Authors · 2021-08-10
> **Reply to reviewer y9u4**
>
> Thank you for your recognition of the results and writing of our paper. We will revise our paper carefully according to your comments. Here are our answers to your comments and questions:
>
> Q1: The construction for the kernel generator is described in detail in page 5. But I do not see any clear construction for the encoder . It seems to me from Figure 1 that the encoder is chosen to be a linear map.
> If yes, is there any reason for choosing to be a linear map rather than an MLP?
>
> A1: Yes, we choose the V to be the linear map, and we will emphasize the structure of V in the revision. Of course, we can adopt it as MLP. We have tried but it results in more parameters and computations without bringing further improvement.
>
> Q2: An explanation for the relation of E4-layer with G-Conv layer and G-SA layer is needed.
>
> A2:
> (1) G-conv is a linear group equivariant model, while the $E^4$-layer and G-SA is non-linear group equivariant model.
>
> (2) $E^4$-layer decouples the aggregation of information along the $\mathcal{A}$ dimension and spatial dimension into two steps while the other two models finish it in a whole process.
>
> (3) Methods of making use of position information are different. For G-conv in Eqn.5, the position information is encoded by attaching different weights $W(g^{-1}{\tilde{g}})$ for different relative positions $g^{-1}{\tilde{g}}$. For G-SA, the position information is encoded by introducing a position encoding $P_{g^{-1}\tilde{g}}$, and for the $E^4$-layer, the position information is implicitly encoded in the organized output form of our kernel generator.
>
> (4) G-conv is spatial-agnostic while the other two are not.
>
> Q3: In experiments, comparisons of the accuracy of E4-Net with G-CNNs are reasonable. Why do not you compare with G-SA neural nets?
>
> A3: We construct the model p4-SAR18 by replacing the second group convolution layer in each Res-Block of p4-R18 with G-SA layer. We carry out the experiments on CIFAR10 and CIFAR100 in the same setting of p4-$E^4$R18. Results are listed in Table IV. We can see that for p4-SAR18 the computation cost is more expensive and the performance is weaker than standard G-CNN, which is consistent with results shown in [24].
>
> Table IV: Results of G-SA layer on CIFAR.
>
>  | Model | error on CIFAR10 (%) | error on CIFAR100 (%) | params | flops |
>  | --- | --- | --- | --- | --- |
>  | p4-R18 |7.53 | 27.96 | 11M | 2.99G |
>  | p4-$E^4$R18 |6.42 | 26.59 | 5.8M | 1.85G |
>  | p4-SAR18 | 12.8 | 36.3 | 10.8M |3.67G |

---

### Official Review · Reviewer_F1d8 · 2021-07-20

**Rating:** 7
**Confidence:** 4

**Summary:**

This paper introduces a functional abstraction for group-convolutional networks employing first- and second-order features. Based on this functional abstraction, a new functional form for second-order group-convolutional networks is devised, which can be made more parameter-efficient than existing approaches.
Experiments are performed on rotated MNIST and CIFAR, evaluating error, number of parameters and certain ablations.

**Limitations And Societal Impact:**

Limitations are not really discussed.
I do not foresee potential negative societal impact of this work.

**Main Review:**

While I enjoyed reading about the generalization idea, I feel that there is a certain amount of imprecision hampering it.

Some general comments and questions:

- The statement that the authors "propose a generalized framework of equivariant models" could be construed as having found the most general formulation. This is certainly not the case, given higher-order interactions that cannot be framed this way.

- The "spatial-agnostic problem" is not a problem. It is just an architecture choice.

- Most CNNs, and incidentally the network proposed in this paper, are not actually equivariant to pixel shifts due to the pooling

- the formulation of the kernels with neighborhoods \mathcal N(g) is a bit vague. It should be made clear that the precise indexing of the elements of the neighborhood is important if one does not want to resort to permutation invariant functions such as averaging. It is mentioned briefly when incorporating group \mathcal A in the example. A further question would be: It would seem that these neighborhoods need to exhibit some form of transformability as well. Can this be simply characterized? At some points, MLPs are proposed to handle whole neighborhoods. How would these be made to transform correctly?

- The experiments on CIFAR are a bit contrived -- please compare to fully data-augmented CNNs. If this approach is to be shown to be useful, it should be able to perform at the level of data augmentation using the symmetries encoded in the network. Data augmentation may also help in the proposed architecture because of the lack of shift-invariance below aggregated stride size.

- A demonstration of object recognition on ImageNet would truly show this approach to be efficient. This benchmark is also a form of water-shed - many architectures work on CIFAR that do not work on ImageNet

- the symmetry groups chosen are not very interesting. It would have been nice to see some more non-trivial rotation at least.

- style suggestion: s/Besides, //g

**Time Spent Reviewing:**

3

---

> ### Author Response · Authors · 2021-08-10
> **Reply to reviewer F1d8**
>
> Thank you for your enthusiastic suggestions on our work. We will revise our paper carefully according to your comments. Here are our answers to your comments and questions:
>
> Q1: The statement that the authors "propose a generalized framework of equivariant models" could be construed as having found the most general formulation. This is certainly not the case, given higher-order interactions that cannot be framed this way.
>
> A1: Here the generalized framework means that the many previous equivariant networks can be formulated in our framework. Of course, higher interaction can be considered to make the framework more general, which can be further studied in future works. We will better phrase the sentence in the revision.
>
> Q2: The "spatial-agnostic problem" is not a problem. It is just an architecture choice.
>
> A2: Spatial-agnostic, also called content-agnostic or spatial-invariance, is one of the shortcomings of standard convolution networks which is detailedly discussed in [Su et al.(2019)] and [Wu et al(2018)].
> To speak more specifically, consider the gradient of a convolution layer during the training process, different descend directions are applied to minimize the loss at different positions. If the kernel is spatially shared, the loss gradients at each position will be globally pooled to train the kernel, leading to the sub-optimal kernel learning that the global gradient is zero while local gradients are non-zero. From this point of view, we can choose position-specific kernels to alleviate this problem.
>
> References:
> [Su et al.(2019)] Su, Hang, et al. "Pixel-adaptive convolutional neural networks." IEEE/CVF Conference on Computer Vision and Pattern Recognition. 2019.
>
> [Wu et al(2018)] Wu, Jialin, et al. "Dynamic filtering with large sampling field for convnets." European Conference on Computer Vision (ECCV). 2018.
>
> Q3: Most CNNs, and incidentally the network proposed in this paper, are not actually equivariant to pixel shifts due to the pooling.
>
> A3: The shift equivariance problem of the pooling layer indeed exists in CNNs-like network architectures, and there have been a lot of works attempting to deal with this issue, e.g., [Zhang 2019]. However, our paper mainly focuses on designing an alternative for group equivariant convolutional layers, so previous works on solving the problem of the pooling layer can be seamlessly incorporated into our network.
>
> Reference:
> [Zhang 2019] Zhang, Richard. "Making convolutional networks shift-invariant again." International Conference on Machine Learning. 2019.
>
> Q4: The formulation of the kernels with neighborhoods $\mathcal N(g)$ is a bit vague. It should be made clear that the precise indexing of the elements of the neighborhood is important if one does not want to resort to permutation invariant functions such as averaging. It is mentioned briefly when incorporating group $\mathcal{A}$ in the example. A further question would be: It would seem that these neighborhoods need to exhibit some form of transformability as well. Can this be simply characterized? At some points, MLPs are proposed to handle whole neighborhoods. How would these be made to transform correctly?
>
> A4: In lines 172-175 of the paper, we have defined $\mathcal{N}(g)=\\{gg^{\prime}|g^{\prime}\in\mathcal{N}(e)\\}$ mathematically. Intuitively, we require the neighborhood of each group element to share the same relative position, just as the sliding windows used in the convolution operation. From our perspective, this exactly characterizes the “some form of transformability of neighborhood”. To make it clearer, in Eqn.12, there are two types of neighborhood $\mathcal{N}_1$ and $\mathcal{N}_2$. The $\mathcal{N}_1$ is used in the aggregation step of Eqn.12. As the summation does not rely on the ordering, the ordering of the elements in $\mathcal{N}_1$ is not necessary. The $\mathcal{N}_2$ is used to concatenate the features. The ordering here is important which is decided by the relative position of each neighborhood element, for example, $i\in \mathcal{N}_2(g)$ is decided by $g^{-1}i$. In practice, we just need to align the features to the right ordering (just like the method in Section7.1 of [5] which precomputes an indices permutation), before feeding them into MLPs.
>
> Q5: The experiments on CIFAR are a bit contrived -- please compare to fully data-augmented CNNs. If this approach is to be shown to be useful, it should be able to perform at the level of data augmentation using the symmetries encoded in the network. Data augmentation may also help in the proposed architecture because of the lack of shift-invariance below aggregated stride size.
>
> A5: Our motivation for reporting the results on CIFAR without data augmentation in our paper is to highlight the data efficiency property of our model.
> We conduct additional experiments on CIFAR with data augmentation and report the results in Table I. As we can see, under the setting using data augmentation, $E^4$-Net still significantly outperform G-CNNs.
>
> Table I: Results on CIFAR with data augmentation.
>
>  | Model | error on CIFAR10 (%) | error CIFAR100 (%) |
>  | --- | --- | --- |
>  | p4-R18 | 4.4 | 19.54 |
>  |p4-$E^4$R18 | 3.9 | 18.76 |
>
> Q6: A demonstration of object recognition on ImageNet would truly show this approach to be efficient. This benchmark is also a form of water-shed - many architectures work on CIFAR that do not work on ImageNet.
>
> A6: We have conducted the experiments on ImageNet to demonstrate the performance of our model. We choose R18, p4-R18 and p4-$E^4$R18 which are described in section 5.2, except that the last fully connected layer are replaced to deal with classification of 1000 category. In the experiments, we adopt commonly used data augmentation as in [10] and train all these models for 120 epochs utilizing the Stochastic Gradient Descent (SGD) optimizer with the momentum of 0.9 and the weight decay of 0.0001. The learning rate initiates from 0.3 and gradually approaches zero following a half-cosine function shaped schedule. No training tricks are adopted. The results are listed in Table II. Our model significantly outperforms G-CNNs with smaller model size on the ImageNet which is consistent with results on CIFAR.
>
> Table II: Results on ImageNet.
>
> | Model | top1 error(%) | top5 error(%) |
> | --- | --- | --- |
>  | R18 | 28.71 | 9.8 |
> | p4-R18 | 25.3 | 7.67|
> | p4-$E^4$R18 | 23.8 |6.9 |
>
> Q7: The symmetry groups chosen are not very interesting. It would have been nice to see some more non-trivial rotation at least.
>
> A7: Here, we further consider the p8 group, which is composed of planar rotation of angles that are multiples of $\pi/4$. Including the p8 group, our $E^4$ layer is more or less the same as the p4 case. We conduct experiments on CIFAR with data augmentation. The network architecture and other training settings are kept the same as section 5.2 of our paper. Results are listed in Table III. As shown in the table, incorporating more rotational symmetries further improve performance of our model.
>
> Table III: Results on p8 group.
>
> | Model | error on CIFAR10 (%) | error on CIFAR100 (%) |
> | --- | --- | --- |
> | p8-R18 | 4.2 | 18.7 |
> | p8-$E^4$R18 | 3.6 | 17.9 |
>
>
> Limitation: In Eqn.12, we just give a special case of the function form $H_\hat{g}$, and further exploration in the space of function $H_\hat{g}$ is in demand. Our work does not provide with a theoretical guide for designing function $H_\hat{g}$  that makes the model more powerful.

---

> ### Author Response · Authors · 2021-08-25
> **Follow up messages**
>
> Thanks again for the valuable comments. Please let us know if anything is unclear in our response. We truly appreciate this opportunity to improve our work and shall be most grateful for any feedback you could give to us.

---

> ### Author Response · Authors · 2021-08-30
> **We'd like to hear your opinions on our rebuttal**
>
> Dear Reviewer,
> We have written a long rebuttal, including adding new experiments, to address your concerns. We wonder whether our rebuttal is adequate? If so, would you please update your score? We know that you may be busy. Thank you for your additional time!

---

> ### Comment · Reviewer_F1d8 · 2021-09-02
>
> Please excuse the late reply.
>
> I am impressed by the amount of extra work done during this review response phase and find that my comments have been adequately addressed. I raise my score to 7.

---

> > ### Author Response · Authors · 2021-09-02
> > **Thanks for your feedback**
> >
> > Dear reviewer, thanks for your nice feedback and updating your score.
> >
> > Best regards, The Authors

---

### Decision · Program_Chairs · 2021-09-27

**Decision:**

Accept (Poster)

**Comment:**

The authors have written a very convincing rebuttal that leans me toward acceptance of this paper.